# Open Source Repository and Online Calculator of Prediction Models for Diagnosis and Prognosis in Oncology

**DOI:** 10.3390/biomedicines10112679

**Published:** 2022-10-23

**Authors:** Iva Halilaj, Cary Oberije, Avishek Chatterjee, Yvonka van Wijk, Nastaran Mohammadian Rad, Prabash Galganebanduge, Elizaveta Lavrova, Sergey Primakov, Yousif Widaatalla, Anke Wind, Philippe Lambin

**Affiliations:** 1The D-Lab, Department of Precision Medicine, GROW-School for Oncology, Maastricht University, 6211 LK Maastricht, The Netherlands; 2Health Innovation Ventures, 6229 EV Maastricht, The Netherlands

**Keywords:** web application, open-source software, prediction models, prediction tool, decision support

## Abstract

(1) Background: The main aim was to develop a prototype application that would serve as an open-source repository for a curated subset of predictive and prognostic models regarding oncology, and provide a user-friendly interface for the included models to allow online calculation. The focus of the application is on providing physicians and health professionals with patient-specific information regarding treatment plans, survival rates, and side effects for different expected treatments. (2) Methods: The primarily used models were the ones developed by our research group in the past. This selection was completed by a number of models, addressing the same cancer types but focusing on other outcomes that were selected based on a literature search in PubMed and Medline databases. All selected models were publicly available and had been validated TRIPOD (Transparent Reporting of studies on prediction models for Individual Prognosis Or Diagnosis) type 3 or 2b. (3) Results: The open source repository currently incorporates 18 models from different research groups, evaluated on datasets from different countries. Model types included logistic regression, Cox regression, and recursive partition analysis (decision trees). (4) Conclusions: An application was developed to enable physicians to complement their clinical judgment with user-friendly patient-specific predictions using models that have received internal/external validation. Additionally, this platform enables researchers to display their work, enhancing the use and exposure of their models.

## 1. Introduction

One of the most challenging areas of modern medicine is oncology [1]. The decision-making process for the best treatment plan is now more difficult than ever because of the inherent heterogeneity of cancer types, patients, and the ever-growing range of available treatments [2]. To choose the “optimal” course of treatment, clinicians must consider evidence from clinical trials, continuing research, their own professional expertise, national guidelines, and the values of the patient [3]. Predictive modeling is becoming a key knowledge-based tool in the healthcare field [4]. The popularity of predictive modeling arises from advancements in a variety of areas, including the availability of health data from electronic health records and databases, a better understanding of causal or statistical predictors of health, disease processes, and multifactorial models of poor health, and advances in nonlinear computer models based on artificial intelligence or neural networks [5]. These new computer-based modeling techniques are gaining credibility in therapeutic settings [6]. However, the current understanding of how so-called machine intelligence will evolve, and thus how current relatively sophisticated predictive models will evolve in response to advances in technology, is difficult to predict [7]. What is known is that despite the fear of the black-box nature of certain AI models, with the introduction of various nomograms dealing with predictive models in oncology, their simplicity and ease of use is helping to cross the bridge of academic/research use to clinical adoption [8,9,10].

There was a need for a public repository of predictive models, and a thorough search was conducted to identify existing applications. However, as the search did not yield useful results, it became our goal to develop a web-based application that would compile and archive publicly available, validated models [11]. The doctors who use such an application would be able to quickly assess the benefits and drawbacks of the chosen models that passed the quality controls. They could securely rely on the application to calculate the output of such models by providing the inputs in a specific wizard rather than designing their own implementation or researching for an appropriate implementation on their own.

The advantages for medical researchers include enhancing the exposure of their model, which should encourage usage and citations, and assisting them in generalizing their models by enabling the models to be evaluated by research groups other than the ones that generated them TRIPOD 4 (Transparent Reporting of studies on prediction models for Individual Prognosis Or Diagnosis) [12]. This paper will describe the foundational work that led to the creation of this application prototype.

## 2. Materials and Methods

The National Center for Biotechnology Information’s (NCBI) PubMed database and Medline papers from December 2016 to March 2022 were reviewed. The following search terms were utilized to find publications that were pertinent: “validated prognostic models in oncology”, “novel diagnostic tools for cancer with external validation”, “lung, prostate, head and neck predictive models”, and “machine-learning & deep learning models in oncology”. Additionally, all the models included were validated (TRIPOD type 3 or 2b) or open-source repositories or peer-reviewed articles. Since this is not a systematic review, no more than five models for each disease (brain, head and neck, lung, prostate, esophagus, rectum, and endometrium) were selected and translated onto a user interface. Figure 1 illustrates the steps taken from the stage of doing a literature search to that of publishing the results online.

In order to assess the category of the models from the studies, every paper was tested for its compliance with the TRIPOD classification, as shown in Figure 2.

The process for extracting the coefficients from a nomogram and implementing them into a web user interface will be described in the following paragraphs.

### 2.1. Obtaining Model Coefficients from a Nomogram

A large number of predictive models with respect to treatment outcomes (cure rate, cancer recurrence, survival, toxicity) are available in the literature. Regression models are frequently published as nomograms in order to make them easier for medical specialists to read and interpret, often without disclosing the model coefficients. A straightforward technique was utilized to extract the coefficients from nomograms in order to publish the models uniformly on the application. The papers in which nomograms are published will usually report the type of model (e.g., logistic regression); otherwise, this can be determined by examining the part of the nomogram where the total score is mapped to a probability. In the nomogram, numbers of points are assigned to different inputs, for example, female might have more points than male. These points are summed over all the inputs, resulting in a nomogram score. This score is then related to a probability of an outcome. The relationship between the total score and the probability is described by the equation for the model. To obtain an equation describing the nomogram, the total score of the nomogram and the probability are read by digitizing the nomogram, and a linear fit is made using the function describing the model, e.g., Equation (1):(1)logit(p)=a+b×Score
where p is the probability, a is the intercept, and is b the scaling factor. Once the relationship is determined between the model parameters and the total score, the coefficients can be calculated. In order to be as precise as possible when reading the nomograms, any graph digitalization software can be used. An example from one of the utilized models is used to illustrate this approach, and it is represented in Figure 3 below [3].

In this instance, a logistic regression model is used to determine the link between the coefficients and the likelihood; as a result, the association between the nomogram score (total points) and the standard logistic distribution of the probability is linear, from which the slope and intercept can be determined (Figure 4).

Next, the relationship between the parameters and the nomogram score can be extracted by reading the nomogram, and the coefficient can be obtained by multiplying by the scaling factor (0.307), as shown in Table 1.

The probability P can now be obtained through Equations (2) and (3):(2)$$y=-1.47+x_{1}\times 0.72+x_{2}\times 0.34+x_{3}\times 0.30+x_{4}\times 0.61$$
(3)$$P=\frac{1}{1+e^{-y}}$$
where $x_{1}$ to $x_{4}$ are 0 or 1 (see Table 1). It is possible to perform a check on how well this equation describes the nomogram by filling in some values into the equation and reading the nomogram to test if the answers are the same. For the example above, you could enter a patient with squamous cell carcinoma, differentiation grade 2, male gender, and a tumor with T-stage 2. Totaling the points (100, 0, 0, and 85) results in a nomogram score of 185, which correlates to a probability of just over 45% when reading the nomogram. Filling this in the equations mentioned above ($x_{1}=1,\;x_{2}=0,\;x_{3}=0,\;x_{4}=1$) results in P=0.465. The exact performance of the equation is often influenced by rounding factors or reading errors.

### 2.2. Application Development Process

In order to automate the self-import process of data models and to create a central model repository, a web-based application has been created that can be accessed by researchers from around the world, as shown in Figure 5.

The development of the web application was done using Python 3.6. and the Django v3.1 web framework. One of the main reasons for choosing Python as the main programming language for this application is because it would allow easier integration of existing research that is based on Python. Furthermore, Django is a highly capable web framework that provides a rich feature set for easy implementation and management of a web app, such as built-in support for a database and an admin console.

For the back-end of the application, a PostgresSQL database was used to store the details related to the model, such as the author of the model, intended end users, and how the model has been developed. In addition, there are stored the parameters (i.e., age, volume, sphere diameter, type of treatment, stage) of each model to dynamically generate the UI in order to facilitate the testing of the model. For the development of the front end of the web application, technologies such as Javascript, Bootstrap 3, HTML5, and CSS3 were used alongside the Django web framework, as shown in Figure 6. The materials and supplements contain comprehensive information about the database (Appendix A).

AI4Cancer is intended for the open-source publishing of prediction models, developed to predict outcomes for cancer patients.

## 3. Results

The developed open-source application is accessible online at https://ai4cancer-ai.herokuapp.com/models-browser# (accessed on 1 April 2022). As shown in Figure 7, it will serve as a repository for published AI prediction models that address all aspects of various cancer types and stages, including diagnosis, prognosis (patient therapy, risk stratification), and follow-up (treatment result, consequences).

Every model that is displayed has an explanation of the technique and clinical datasets that were used to create and validate the model. The limitations of each model are explicitly mentioned. The developed models are lung cancer, rectum, head and neck cancer, and brain metastases, and they are developed to be used only by physicians, not patients, due to the complexity of cancer treatment decision-making options. There are several different types of models that are used, including linear regression, tumor control probability (TCP), linear-quadratic (LQ), Kaplan–Meier, linear-quadratic biologically effective dose (LQ-BED), Cox, and logistic regression [15,16]. As shown in Table 2, there are now 18 models that have been released and deployed.

For each online model given, doctors can find: (a) the intended use (predicted outcome) of the model, (b) to which patients it applies, (c) the information and the parameters that the doctor must enter, and (d) how the model was developed. The doctors can access the website, select a suitable model, and enter the required information to produce the likelihood of survival. The website’s predictive models follow the same calculations as the models outlined in the academic works on which they are based.

The primary output of this research is a widely applicable application that comprises verified models for various stages, odds of survival, symptoms, and outcomes. The collection of these predictive models will help doctors make decisions.

## 4. Discussion

This application can be considered a “model zoo” for academics and medical professionals who are well-versed in the medical ramifications of various cancer forms. None of the models are intended to replace clinical judgment; rather, they are all intended to advance research. The open-source application is not meant for usage by laypeople without assistance (e.g., patients). It is important to emphasize that this manuscript and application are currently only prototypes. The inclusion of all models that satisfy the selection criteria is not claimed. Similarly, any models that are not available on the platform right now should not be viewed as a cause for concern or as a rejection of their validity.

For heightened patient privacy, models that require medical (DICOM) images are not incorporated, as the pipeline does not currently have the ability to check that such uploaded data was properly anonymized. By working together with outside organizations (such as hospitals) that are in charge of preserving the privacy of such data, such machine learning models will be incorporated. In future iterations of the website, it will also be possible to categorize models according to the outcome they are modeling, as well as divide models according to the kind of input data they need. This will result in a more structured semantic-based organization. When the division of models was done based on the result being modeled, there will be dedicated website pages that compare models created to predict a specific outcome.

The models have not been validated for individual use; but rather, only in clinical cohort research. These models should not be used by patients directly and should only be used by doctors who are knowledgeable about the complexity of cancer types. The main intention of these models is to inform doctors and they should not be used for decision support.

Researchers should view this work as an invitation to use this application, with the ability to hide the code from the user while still providing full functionality. Researchers’ models will be successfully integrated with the aid of the application itself. This will generate synergies that will inevitably speed up oncology-related AI research. Additionally, it will prevent models from being unnoticed and underutilized, which frequently occurs when numerous publications on the same general topic are released quickly.

There are certain limitations to the technique employed to extract regression model coefficients from a nomogram. One limitation is that the resolution of the published model significantly impacts accuracy. Another limitation is that even though the model coefficients may be retrieved, a nomogram cannot be used to determine the standard error for the parameter coefficients [48]. However, the technique may be employed with any nomogram, making it a useful tool.

By encouraging more researchers to submit their work via this application, hopefully, the number of curated models will increase and the application can continue to serve the public. This will be accomplished as a result of an increase in staff via the Optima Grant, who will perform monthly checks for the new models. The three prerequisites are (1) open-access papers with a clear, implementable model description; (2) papers with a minimum of 10 citations; (3) models of TRIPOD 3 and 4.

## 5. Conclusions

The repository accessible at https://ai4cancer-ai.herokuapp.com (accessed on 1 April 2022), in the current prototype level, contains 18 proven machine learning models to assist doctors in making decisions on patient care for a variety of cancer types. Other researchers can make use of this technique for deriving regression coefficients from nomograms. As a result, research teams are being urged to disseminate their models globally.

## Figures and Tables

**Figure 1 biomedicines-10-02679-f001:**
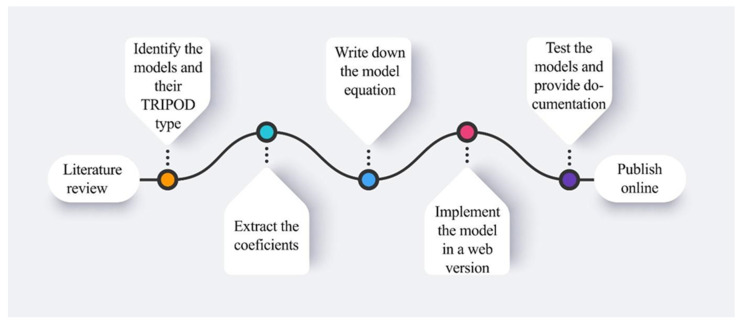
The pipeline from selecting the convenient models until the publishing phase.

**Figure 2 biomedicines-10-02679-f002:**
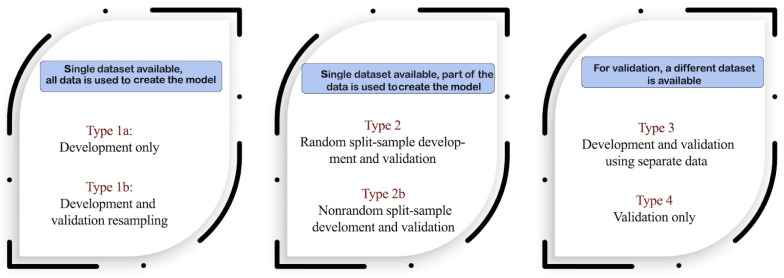
TRIPOD classifications [13].

**Figure 3 biomedicines-10-02679-f003:**
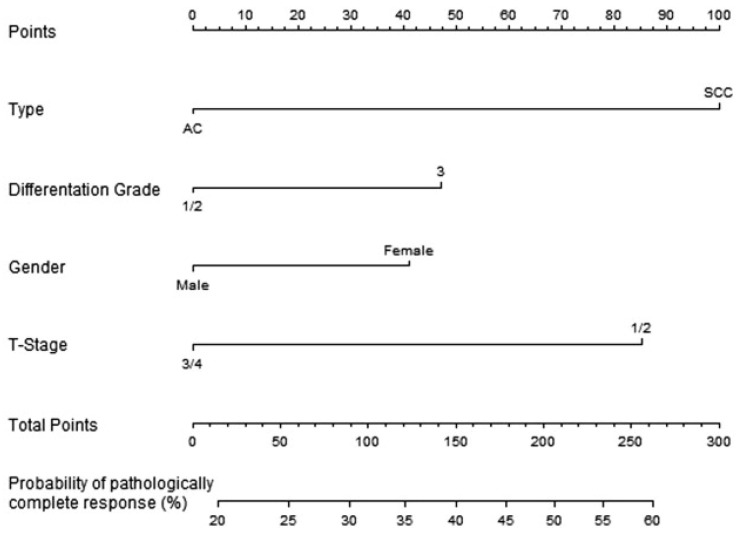
Nomogram for predicting pathologically complete response after neoadjuvant chemoradiotherapy for oesophageal cancer [14].

**Figure 4 biomedicines-10-02679-f004:**
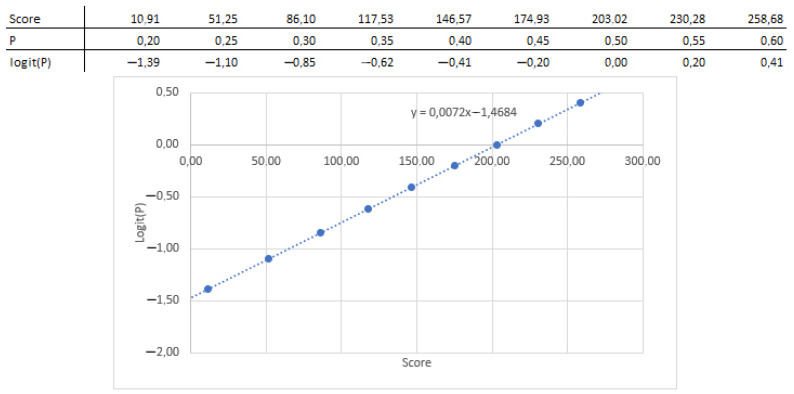
Coefficients and intercept extracted from the nomogram.

**Figure 5 biomedicines-10-02679-f005:**
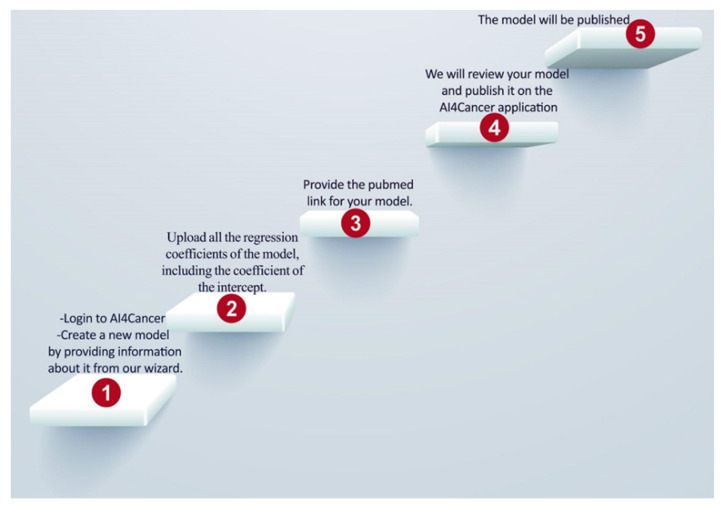
Shows the steps of user-uploading models in the application.

**Figure 6 biomedicines-10-02679-f006:**
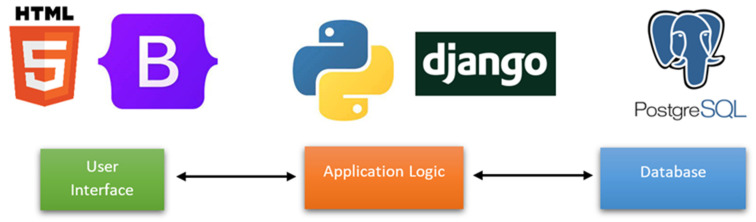
Illustration of the application architecture.

**Figure 7 biomedicines-10-02679-f007:**
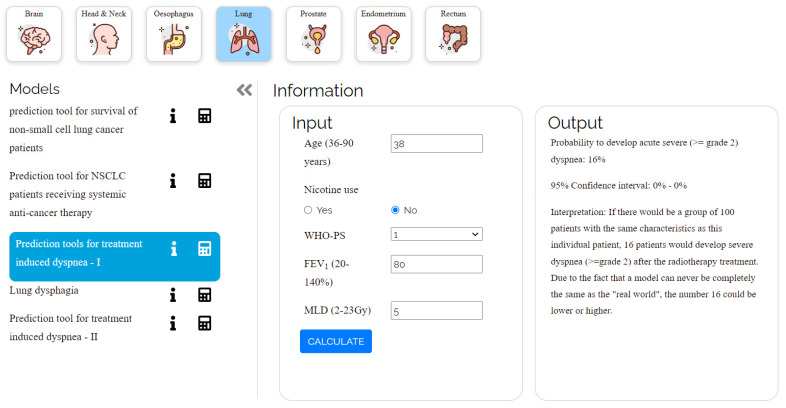
Shows the user interface of a model (lung cancer) from the website.

**Table 1 biomedicines-10-02679-t001:** Point reading for the nomogram.

Parameter	Equation	Nomogram Value	Equation Value	Points	Coefficient
Intercept	-	-	-	-	−1.47
Tumor type	$x_{1}$	Adenocarcinoma	0	0	0
		Squamous cell carcinoma	1	100	0.72
Differentiation grade	$x_{2}$	1 or 2	0	0	0
		3	1	47	0.34
Gender	$x_{3}$	Male	0	0	0
		Female	1	41	0.30
T-stage	$x_{4}$	3 or 4	0	0	0
		1 or 2	1	85	0.61

**Table 2 biomedicines-10-02679-t002:** For every model: cancer type, input features, outcome, cohort type and TRIPOD, and model type.

Model Title	Cancer Type	Input Features	Output	Cohort Type	Tripod Type	Model Type
Brain metastases development after radical treatment of stage III NSCLC patients [17]	Brain	Tumor histology Age	Predicts the development of brain metastasis.	Primary stage III NSCLC brain cancer patients.	2b	Linear regression
2. New distant brain metastases within 1 year after SRS for a maximum of 3 brain metastases [18]	Brain	WHO performance status; Age; Volume of the largest brain metastasis; Number of treated brain metastases	Predicts the probability of occurrence of new distant brain recurrences.	Patients with 1 up to 3 BM treated with SRS.	2b	Univariate logistic regression and Cox regression
3. One-year local control probability after stereotactic radiosurgery for brain metastases [19,20,21,22,23,24,25,26,27,28,29,30]	Brain	Prescribed fraction SRS dose; Number of SRS fractions	Predicts local control probability at 1 year after stereotactic radiosurgery (SRS) for brain metastases (BM).	Patients treated with SRS for BM.	2b	Linear-quadratic BED (LQ-BED), Linear regression, Cox regression
4. BMNSCLC: Early death and long-term survival after stereotactic radiosurgery for 1–3 brain metastases of NSCLC [31]	Brain	Age; Presence of extracranial metastases; WHO performance status; GTV largest metastasis; Volume; Sphere; diameter	Predicts the probability of early death (<3 months) and the probability of long-term survival (>12 months) with prognostic factors for survival.	Patients treated with SRS for 1, 2, or 3 BMs of NSCLC.	3	Multivariate Cox regression
5. Xerostomia after head and neck cancer radiotherapy [32]	Head and Neck	D-mean-contra; D-mean-ipsi	Predicts Xerostomia three months after intensity-modulated radiotherapy (IMRT).	Patients with locally advanced head and neck cancer, eligible for potentially curative loco-regional treatment, and have not been treated for another malignancy.	3	Logistic regression
6. Cost-effectiveness of IMPT versus IMRT [33,34,35]	Head and Neck	Mean dose of ipsilateral parotis; contralateral parotis; pharyngeal constrictor muscle superior; supraglottic area; Willingness to pay per QALY gained	Predicts the most cost-effective treatment.		2b	Multivariate logistic regression
7. Sticky saliva, 6 months after treatment [36]	Head and Neck	Mean dose contralater submandibular gland; Mean dose sublingual glands; Mean dose soft palate	Predicts the probability of sticky saliva 6 months after treatment.	Head and neck cancer (HNC) originating in the oral cavity, oropharynx, larynx, hypopharynx, or nasopharynx; 2. treated with curative intensity-modulated radiotherapy (IMRT) either alone or in combination with chemotherapy or cetuximab; 3. no previous surgery, radiotherapy and/or chemotherapy; 4. no previous malignancies; 5. no distant metastases; 6. planning CT and 3D-dose distributions available in DICOM format; 7. HRQoL assessments available prior to and 6 months after completion of (CH)RT.	2b	Multivariate logistic regression
8. Tube Feeding Dependence, 6 months after treatment [37]	Head and Neck	T-classification; Baseline weight loss; Type of treatment; Mean dose PCM superior; Mean dose PCM inferior; Mean dose contralateral parotid; Mean dose cricopharyngeal muscle	Predicts the probability of tube feeding dependence 6 months after treatment.	Curative radiotherapy/chemoradiotherapy for head and neck cancer (HNC) may result in severe acute and late side effects, including tube-feeding dependence.	2b	Multivariable logistic regression
9. Pathological complete response (pCR) [13]	Esophagus	Tumor histology; differentiation grade; Gender; T-stage	This tool predicts pathological complete response.	Patients with histologically proven carcinoma of the esophagus or gastro-esophageal junction, treated with neo-adjuvant chemoradiotherapy (CROSS) followed by surgery.	2b	Logistic regression
10. Prediction tool for survival of non-small cell lung cancer patients [38]	Lung	Gender; WHO-PS; FEV1; Gross tumor volume; Number of nodal stations	Predicts the probability that a patient with non-small cell lung cancer (NSCLC) will be alive at 2 years post-radiotherapy treatment.	NSCLC patients, stage I-IIIB.	3	Kaplan–Meier and Cox regression
11. Prediction tool for NSCLC patients receiving systemic anti-cancer therapy [39]	Lung	Age; Gender; Performance status; Status; Income deprivation; Previous treatment given; BMI	Predicts the probability of 30-day mortality.	NSCLC patients receiving systemic anti-cancer therapies (SACT).	3	Logistic regression
12. Prediction tools for treatment-induced dyspnea—I [40,41]	Lung	Age; Nicotine use; WHO-PS; FEV1; MLD	Predicts the probability of developing acute severe (≥grade 2) dyspnea: %.	NSCLC patients, stage I-IIIB and SCLC patients.	3	Multivariate logistic regression
13. Lung dysphagia [42]	Lung	Gender; Age; OTT; Mean esophagus dose; Max esophagus dose; Chemotherapy; WHO-PS	Predicts the probability of developing dysphagia ≥ grade 3: %.	NSCLC stage I-IIIB as well as SCLC patients with limited disease.	3	Multivariable logistic regression
14. Prediction tool for treatment-induced dyspnea—II [43]	Lung	Baseline dyspnea score* at the start of R(CH)T; Cardiac comorbidity; Tumor location; Forced expiratory volume; Sequential chemotherapy	Predicts the probability of dyspnea ≥ 2 within 6 months after the start of R(CH)T: %.	NSCLC patients, stage I-IIIB. Patients have to be treated with high-dose conformal radiotherapy alone (≤3 Gy per fraction) or high-dose conformal radiotherapy combined with chemotherapy (sequential or concurrent).	3	Univariate and multivariate logistic regression
15. ProRaDS: Prostatectomy versus radiotherapy decision support system [12]	Prostate	Age; BMI; Diabetes; Hemorrhoids; Uretra; Pre-treatment erectile function; PSA level; T-Stage; Primary Gleason score; No. of positive biopsy cores; No. of negative biopsy cores; ASA score; Anticoagulants; Nerve-sparing surgery; Androgen deprivation therapy; ADT length; Prior abdominal surgery; Irradiation of pelvic nodes; Mean trigone dose; Mean rectum dose; Rectum volume;	Compares the probability of biochemical failure, and the probability of developing chronic erectile dysfunction, urinary incontinence, or rectal bleeding for either prostatectomy or external beam radiotherapy for the treatment of prostate cancer.	Low- to intermediate-risk prostate cancer patients eligible for primary treatment with either external beam radiotherapy or prostatectomy.	3	Markov model
16. Tumor control probability (TCP) [44,45]	Prostate	Total dose D (45.0–82.8 Gy); Fractional dose d (1.2–10 Gy); Modality type; Risk group	Predicts the chance of 5-year biological no evidence of disease (5-year bNED).	External beam radiotherapy prostate cancer patients.	3	TCP, linear-quadratic (LQ)
17. Recurrences and survival (endometrium) [46]	Endometrium	Age; FIGO histological grade; Myometrial invasion depth; Vascular invasion; Radiotherapy	Predicts the probability that an endometrial cancer patient will have one of the following events within 5 years of follow-up: loco-regional recurrence (LRR), distant recurrence (DR), relapse or death (disease-free survival, DFS), and death (overall survival, OS).	Endometrium cancer patients.	3	Cox proportional hazards, Cox regression
18. Pathologic complete response of locally advanced rectal cancer patients [47]	Rectum	Tumor length (2.0–15.0 cm); SUVmax-pre (1.0–25.0); SUVmax-post (1.0–25.0)	Predicts the probability that a patient with locally advanced rectal cancer (LARC) will have a pathologic complete response after long course chemoradiotherapy (CRT) and surgery.	Patients with locally advanced rectal cancer (LARC).	3	Logistic regression

## Data Availability

Not applicable.

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
