# Peer review of "Open Source Repository and Online Calculator of Prediction Models for Diagnosis and Prognosis in Oncology"

_biomedicines, 2022, doi:10.3390/biomedicines10112679_

Round 1

Reviewer 1 Report

I have found that this is an immensely interesting and relevant article, the authors should be praised for their implementation. Keep on with this model and find a way how to inform the oncologist to use your site.

Author Response

Dear reviewer!

Thank you for favorable your review.

Thank you!

Iva Halilaj

Reviewer 2 Report

The manuscript "Open-Source Repository and Online Calculator ..." by Iva Halilaj and others is interesting and potentially very important.  It was submitted as a technical note and simply describes the foundational work that led to the creation of an open-source repository for predictive and prognostic models regarding oncology.  The prototype has already been launched, April of this year, and hopefully it will prove to be successful.  After carefully reading through it twice I could not find any serious content problems although I think it could be improved by defining some of the excessive number of abbreviations.  Not all readers will be good, up-to-date oncologists that don't need some definition reminders; so, I would suggest perhaps adding a pull-down menu of abbreviation definitions to the application.  It would be easy to do and might avoid confusion among assistants in charge of data entry into the applications.  A few equations are also in need of clarification. A few line-by-line suggestions follow:

Line 79 – Move the TRIPOD definition to where it is first used on line 60.

Lines 94-98 – Five lines are not enough to describe the key equation used.  How is the function describing the model calculated? How is it "fit over the total score of the nomogram with a scaling factor and an intercept"? How is the score of the nomogram calculated?  An example is provided on Figures 3 and 4 and Table 1 but there is still some unnecessary confusion.  

Figure 4 – Extend the line so that it actually intersects

Table 1.  - Exactly how are X1, X2, X3, and X4 quantified?

Figure 7. -  Output caption – Delete "... but 0 is the most likely value. The 95% confidence interval indicates that the value will lie between 0 and 0 in 95% of the times that you would have such a group of 100 patients," It serves no purpose and just creates confusion.

Line 163 – Add a space between lines 163 and 164.

Line 209 – How will it prevent models from being unnoticed? Do you plan to hire staff that will monitor about 100 medical journals to ensure that the repository will include all good models? What about potentially important models that are published in fringe journals?

Author Response

Dear reviewer!

Thank you for your favorable review, as for your feedback:

Not all readers will be good, up-to-date oncologists that don't need some definition reminders; so, I would suggest perhaps adding a pull-down menu of abbreviation definitions to the application.  It would be easy to do and might avoid confusion among assistants in charge of data entry into the applications.  A few equations are also in need of clarification. A few line-by-line suggestions follow:

We do understand your concern; we are updating the website this week with all the abbreviations.

Line 79 – Move the TRIPOD definition to where it is first used on line 60.

The definition has been moved in Line 63. Thank You.

Lines 94-98 – Five lines are not enough to describe the key equation used.  How is the function describing the model calculated? How is it "fit over the total score of the nomogram with a scaling factor and an intercept"? How is the score of the nomogram calculated?  An example is provided on Figures 3 and 4 and Table 1 but there is still some unnecessary confusion.  

This is a valid concern as we rely too much on our example for clarity. Extra descriptions were added to clarify earlier in the text, lines 95-104.

Figure 4 – Extend the line so that it actually intersects

The fit line was extended in both directions.

Table 1.  - Exactly how are X1, X2, X3, and X4 quantified?

This was indeed left very unclear. The quantity of X1 etc is related to but not quantified by the inputs of the nomogram. For this reason we added an extra column to Table 1 to clarify this issue.

Figure 7. -  Output caption – Delete "... but 0 is the most likely value. The 95% confidence interval indicates that the value will lie between 0 and 0 in 95% of the times that you would have such a group of 100 patients," It serves no purpose and just creates confusion.

The figure has been replaced based on this valid comment.

Line 163 – Add a space between lines 163 and 164.

Done

Line 209 – How will it prevent models from being unnoticed? Do you plan to hire staff that will monitor about 100 medical journals to ensure that the repository will include all good models? What about potentially important models that are published in fringe journals?

We do understand your concern but we will extend staff via IMI-OPTIMA n° 101034347 Grant to do monthly checks for the new models based on selection criteria: 1) open access papers with explicit model description (implementable); 2)based on minimum citation count (min 10), 3) with preference with TRIPOD  3 and 4. We added this in lines 236-239.

Thank you!

Iva Halilaj

Reviewer 3 Report

1) Please write the whole paper in passive voice. Dont use I, We and son on.

2) Write a single para by highlighting the research gap that you found and write your contribution at the end of Introduction section.

3) Put all equation numbers.

4) In Figure 4, the font size should be aligned with the other part of the image.  

5) How did you analysis the performance of your model calculator?

6) Authors need to explain the backend database structure in detail.

6) Read and cite the recent quality paper :  1) Prediction Models for Prognosis of Cervical Cancer: Systematic Review and Critical Appraisal (2021). 2)  Reporting of prognostic clinical prediction models based on machine learning methods in oncology needs to be improved (2021). 3) LungNet: A Hybrid Deep-CNN Model for Lung Cancer Diagnosis Using CT and Wearable Sensor-based Medical IoT Data (2021). 4) Artificial intelligence in oncology: Path to implementation (2021). 5) A classification of MRI brain tumor based on two stage feature level ensemble of deep CNN models (2022).

Author Response

Dear reviewer!

Thank you for your review, as for your feedback:

1)Please write the whole paper in passive voice. Dont use I, We and son on.

This is a valid concern, we have re-written the paper in past tense. Thank you.

2) Write a single para by highlighting the research gap that you found and write your contribution at the end of Introduction section.

This was indeed left unclear, therefore an paragraph was added to highlight the research gap, lines 51-54.

3) Put all equation numbers.

Equation numbers were added.

4) In Figure 4, the font size should be aligned with the other part of the image.  

The figure 4 has been updated.

5) How did you analysis the performance of your model calculator?

This is a valid concern. We make up 5 fictitious people and we use a nomogram to calculate and compare the output and check if they agree or not. In addition, a check is always done to assess any errors caused by rounding or reading uncertainty. To address this, we added a short section on checking the model, lines 95-104.

6) Authors need to explain the backend database structure in detail.

 In the materials and supplements comprehensive information about the database has been added as attached below.

7) Read and cite the recent quality paper :  1) Prediction Models for Prognosis of Cervical Cancer: Systematic Review and Critical Appraisal (2021). 2)  Reporting of prognostic clinical prediction models based on machine learning methods in oncology needs to be improved (2021). 3) LungNet: A Hybrid Deep-CNN Model for Lung Cancer Diagnosis Using CT and Wearable Sensor-based Medical IoT Data (2021). 4) Artificial intelligence in oncology: Path to implementation (2021). 5) A classification of MRI brain tumor based on two stage feature level ensemble of deep CNN models (2022).

This is a very valuable point for this paper. All these paper have been cited, thank you.

Thank you!

Iva Halilaj

Round 2

Reviewer 3 Report

The authors have addressed all the comments successfully